# Prenatal Exposure to Δ9-Tetrahydrocannabinol Affects Hippocampus-Related Cognitive Functions in the Adolescent Rat Offspring: Focus on Specific Markers of Neuroplasticity

**DOI:** 10.3390/pharmaceutics15020692

**Published:** 2023-02-17

**Authors:** Valentina Castelli, Gianluca Lavanco, Salvatore Feo, Cesare D’Amico, Vincenzo Micale, Martin Kuchar, Fulvio Plescia, Anna Brancato, Carla Cannizzaro

**Affiliations:** 1Department of Biomedicine, Neuroscience and Advanced Diagnostics, University of Palermo, 90127 Palermo, Italy; 2Department of Health Promotion, Mother and Child Care, Internal Medicine and Medical Specialties of Excellence “G. D’Alessandro”, University of Palermo, 90127 Palermo, Italy; 3Department of Biological, Chemical and Pharmaceutical Sciences and Technologies, University of Palermo, 90128 Palermo, Italy; 4Department of Biomedical and Biotechnological Sciences, University of Catania, 95123 Catania, Italy; 5Forensic Laboratory of Biologically Active Substances, Department of Chemistry of Natural Compounds, University of Chemistry and Technology, 166 28 Prague, Czech Republic; 6Psychedelic Research Centre, National Institute of Mental Health, Topolová 748, 250 67 Klecany, Czech Republic

**Keywords:** prenatal THC exposure, adolescent rat offspring, spatial learning and memory, hippocampal excitatory plasticity, CB1R expression

## Abstract

Previous evidence suggests that prenatal exposure to THC (pTHC) derails the neurodevelopmental trajectories towards a vulnerable phenotype for impaired emotional regulation and limbic memory. Here we aimed to investigate pTHC effect on hippocampus-related cognitive functions and markers of neuroplasticity in adolescent male offspring. Wistar rats were exposed to THC (2 mg/kg) from gestational day 5 to 20 and tested for spatial memory, object recognition memory and reversal learning in the reinforce-motivated Can test and in the aversion-driven Barnes maze test; locomotor activity and exploration, anxiety-like behaviour, and response to natural reward were assessed in the open field, elevated plus maze, and sucrose preference tests, respectively. The gene expression levels of NMDA NR1-2A subunits, mGluR5, and their respective scaffold proteins PSD95 and Homer1, as well as CB1R and the neuromodulatory protein HINT1, were measured in the hippocampus. pTHC offspring exhibited deficits in spatial and object recognition memory and reversal learning, increased locomotor activity, increased NR1-, decreased NR2A- and PSD95-, increased mGluR5- and Homer1-, and augmented CB1R- and HINT1-hippocampal mRNA levels. Our data shows that pTHC is associated with specific impairment in spatial cognitive processing and effectors of hippocampal neuroplasticity and suggests novel targets for future pharmacological challenges.

## 1. Introduction

Cannabis is the illicit drug most commonly used by pregnant women who intend to relieve nausea, vomiting, and sleep disturbances [1,2]. This global trend is fuelled by misperceiving cannabis as a harmless self-medicating effective strategy [3,4,5]. Notably, Δ9-tetra-hydrocannabinol (THC), cannabis main psychoactive component, can interact with the endocannabinoidergic system (ECS) in the brain since ontogenesis through the cannabinoid type 1 receptor (CB1Rs) [6,7]. Endocannabinoids (eCBs), by the activation of CB1Rs, play a critical regulatory role in brain maturation throughout all developmental stages, from neural stem-cell differentiation and synaptogenesis to interneurones formation and connectivity [8,9]. Indeed, because of the large distribution of CB1R in functionally relevant brain areas, the ECS contributes to modulating emotional, motivational, and cognitive responses [10,11,12]. In particular, CB1Rs in the hippocampus are localized on GABAergic interneurones and glutamatergic axon terminals, where the ECS regulates synaptic plasticity underlying learning and memory functions [13,14,15,16]. Besides regulating short-term forms of synaptic plasticity (depolarization-induced suppression of inhibition or excitation (DSI/DSE) and metabotropic-induced suppression of inhibition or excitation (MSI/MSE)) in the hippocampus [17,18,19], eCBs facilitate conventional forms of long-term synaptic plasticity [14]. In detail, eCBs’ release following the activation of ionotropic glutamate N-methyl-d-aspartate receptors (NMDARs) and the group I metabotropic glutamate receptors 5 (mGluRs5) promotes long-term potentiation (LTP) and mediates long-term depression (eCBs-LTD) [20,21].

A growing body of literature has shown that the optimal interplay between ECS and the glutamatergic effectors promotes functional synaptic plasticity and memory formation [22,23,24]. The NMDARs are heteromeric complexes in which the combination of NR1 with different NR2 subunits confers functional diversity and uniqueness in electrophysiological and pharmacological properties [25,26]. The NR1 subunits are mandatory components of the NMDARs essential for ion selectivity and agonist binding of the NMDA channels. The NR2 subunit is mainly responsible for regulating channel gating and Mg^2+^ dependency [27,28]. The efficacy of activity-dependent excitatory signalling during learning and memory formation relies on the activation of NR2A-containing NMDARs [29] and the following interaction with the scaffolding protein post-synaptic density-95 (PSD95), which provides structural stability in the synapse and is intimately involved with LTP induction and maintenance [30,31]. NR1-NR2 specific arrangement in NMDAR is functionally relevant for normal hippocampal learning and memory processing [29].

To maintain NMDARs under regulation, the histidine triad nucleotide-binding protein 1 (HINT1) cooperates with CB1R to buffer NR1 excessive expression and the related glutamatergic hyperactivation [32,33]. Notably, NMDAR-mediated excessive excitatory signalling, through the suppression of GABAergic inhibition, was shown to be the substrate for the memory impairment induced by cannabinoid exposure [34,35]. On the other hand, the concurrent activation of mGluR5, which is coupled with the functional rearrangement of scaffolding Homer proteins, by recruiting the synthetic enzyme, promotes the formation of the eCBs responsible for the eCBs-mediated LTD, in both excitatory and inhibitory synapses [36,37,38,39]. Interestingly, activation of mGluR5 stimulates endocannabinoid synthesis, by both increasing intracellular Ca^2+^ [37] and by interacting, through the Homer proteins, with the two key proteins responsible for 2-Arachidonoylglycerol synthesis: phospholipase C β, which is activated by mGluR5, and diacylglycerol lipase-α. The formation of this complex enables the rapid formation of 2-AG following mGluR5 stimulation [40].

Given the role of ECS in the developing brain and, in particular, in the shaping of neuronal connectivity and circuit functionality in the hippocampus [41], it is not surprising that the supraphysiological impact on the ECS by cannabis exposure during critical periods of development could change the normal trajectory of cognitive processing. Indeed, human studies on cannabis exposure during vulnerable epochs of development have reported significant deficits in sustained attention, information processing, learning and memory, and cognitive flexibility in children [42]. However, a plethora of intervening factors introduces considerable variations in clinical observation, making it challenging to reach consistent evidence. In this scenario, the animal model of prenatal THC exposure allows for controlling the confounding variables and the parameters, such as dosage, the timing of exposure, and genetic factors [43]. Our previous evidence [44] suggests that prenatal exposure to THC can lead to the derailment of the neurodevelopmental trajectories towards a vulnerable phenotype for impaired emotional regulation and limbic memory. Those behavioural outcomes were associated with a dysregulation in inhibitory signalling and PSD make-up in mesocorticolimbic regions in adolescent male rat offspring. Indeed, adolescence represents a vulnerable time when the consequences of perinatal insults can arise after exposure to critical environmental stimuli, according to the 2-hit hypothesis, thus playing a major role in the onset of neuropsychological disturbances later in life [45,46]. Given this background, it is reasonable to hypothesize that the manipulation of the ECS by THC exposure during gestation may jeopardize the hippocampal neurodevelopment, thus affecting learning and memory processes and the underpinning molecular players of synaptic plasticity [47].

Thus, here we aim to investigate the putative association between prenatal THC exposure and the occurrence of impairment in hippocampus-related cognitive functions, such as spatial- and object-recognition memory, in adolescent male rat offspring. Moreover, this study investigates the putative occurrence of changes in the expression of critical determinants of hippocampal synaptic plasticity. In detail, we have measured the expression of NMDAR NR1-and NR2A subunits and mGluR5, as well as their respective scaffolding proteins, PSD95 and Homer1. HINT1 expression has been measured as a functional mediator of neuroadaptation. Eventually, modifications of hippocampal CB1R mRNA have been evaluated.

Our evidence confirms our working hypothesis highlighting the relevance of maternal exposure to THC in the occurrence of perturbations in cognitive functions and effectors of hippocampal neuroplasticity.

## 2. Materials and Methods

### 2.1. Animals and Treatment

Twelve adult female pregnant Wistar rats at GD 4 (200–220 g; Envigo, Milan, Italy) were singly housed in standard cages (40 cm × 60 cm × 20 cm), with ad libitum access to water and food in a temperature- and humidity-controlled room (22 ± 2 °C and 55 ± 5%, respectively) on a 12-h light/dark cycle. They were given daily s.c. injections of vehicle (Veh) or THC (2 mg/kg) from gestational days (GD) 5 to 20. The concentration of THC administered in this study and prior research [44,48] corresponds to mild THC consumption in humans. After weaning, male rats were housed in pairs, and each experimental cohort included one or two independent male rats per each litter of Veh- or THC-treated dams (total number = 43 rats). All experiments were approved by the Italian Ministry of Health (819/2021-PR to Carla Cannizzaro) and conducted following animal protocols approved by the Committee for the Protection and Use of Animals of the University of Palermo, in adherence with the current Italian legislation on animal experimentation (D.L. 26/2014) and the European directives (2010/63/EU) on the care and use of laboratory animals. Every effort was made to minimise the number of animals used and their suffering.

### 2.2. Drugs

THC resin, extracted by the Forensic Laboratory of Biologically Active Substances of the University of Chemistry and Technology of Prague, Czech Republic, (purity (HPCL) > 97%) [49], was dissolved in ethanol at 20% final concentration and then sonicated for 30 min. THC was emulsified in 2% Tween 80 and dissolved in sterile physiological saline. Rat dams were subcutaneously administered THC (2 mg/kg) or Veh (1–2% Tween 80, saline) in a volume of 2 mL/kg from GD 5 until GD 20.

### 2.3. Behavioural Procedures

Male rat offspring, prenatally exposed to either Veh (CTRL) or THC (prenatal THC exposure, pTHC), were tested during adolescence—starting from PND 35 to 46—during the light phase of the light/dark cycle. One cohort from each experimental condition performed the open field test (OFT) to assess locomotor activity and exploration in a novel environment [50]; the elevated plus maze test (EPM), to measure anxiety-like behaviour [51]; and the Barnes maze test—a mild aversive, dry-land based behavioural test developed to study spatial learning and memory retrieval, as well as reversal learning in rodents [52]. One other cohort from the different experimental groups was exposed to the sucrose preference (SP) test to investigate the hedonic reactivity [53] and the Can test—a reinforce-motivated task that allows the assessment of both spatial memory and object-recognition memory [54] (Figure 1). The objects and the apparatus used were cleaned thoroughly with 70% isopropanol, dried with tissue paper, and rinsed again with water at the end of each experimental session. The behaviour of the rats was monitored by an experimenter, and videos were recorded by an automatic video-tracking system, AnyMaze (Stoelting Europe, Dublin, Ireland). The behaviour was then quantified by an experimenter unaware of the experimental groups.

### 2.4. Open Field Test

Locomotor activity and exploration in a novel environment were assessed in the open field test [50]. The rat was introduced into a Plexiglas box (44 × 44 × 20 cm) in a mean light intensity (100 lx) illuminated chamber, for 5 min. The behaviour was tracked and recorded by AnyMaze (Stoelting) and the following parameters were quantified: total distance travelled—TDT, as a measure of locomotor activity—the number of central transitions from the peripheral area to the central part of the arena—NCT, as an index of explorative behaviour [55].

### 2.5. Elevated Plus Maze Test

Anxiety-like behaviour was assessed in the elevated plus maze test (EPM) [51]. Rats were tested in a plus-shaped apparatus made from dark-grey PVC, elevated to a height of 70 cm above the floor, consisting of two opposing arms (50 × 10 cm) closed by 40 cm-high side end-walls (closed arms), and two opposing arms with no walls (open arms). The closed and open arms were connected by an open central area (10 × 10 cm). At the beginning of the test, the rat was placed in the centre of the maze facing one of the open arms, and the behaviour was tracked and recorded for 5 min (Any Maze, Stoelting). Time spent on each arm and the number of entries were recorded. An entry was scored when all four paws entered into every single arm. The percentage of time spent on the open arms/total time spent on open and closed arms (Open arm time %) and the percentage of the number of entries in the open arms/number of entries in the open and closed arms (Open arm entries %) were analysed, as they constituted the primary indices of trait anxiety-like behaviour in rodents. The general activity was measured by the mean of the number of total entries [56].

### 2.6. Sucrose Preference Test

To evaluate the responsivity to natural reinforcing stimuli, the adolescent male rat offspring underwent the sucrose preference test (SP) [53]. Rats were individually housed, and one of their two water bottles was replaced with a 1% sucrose bottle for 24 h. The bottles were then weighed, and sucrose preference was calculated as the percentage of the volume of the sucrose solution consumed divided by the total fluid volume (water plus sucrose) [57].

### 2.7. Can Test

For the evaluation of spatial memory in a reward-facilitated context, rats were tested in the Can test, a reinforce-motivated task [54]. Here, the rat was trained to identify a single water-rewarded can, among a set of seven non-rewarded counterparts, employing a 10-h water deprivation schedule for motivation [58]. Cans were painted in white or left in their imprinted colours, according to the task administered, and put upside down in a Plexiglas arena (100 × 100 × 43 cm). This allowed their indented bottoms to hold water. The cans were placed in a fan-shaped pattern so that the distance from each can to the starting point was 70 cm, and the distance between the cans was 7 cm. A “visit” was recorded when the rat stood on its hind paws and brought its nose up to the level of the top edge of the can. The parameters measured were: activity, i.e., the number of trials on which rats visited at least one can (up to 10 during each experimental session); correct responses, i.e., the number of trials in which the rat visited the rewarded can first, divided by the activity score (up to 1 per each experimental session); reference memory errors, i.e., the first visits to a non-rewarded can on each trial, divided by the activity score (up to 6 per each experimental session); working memory errors, i.e., repeated visits to the same non-rewarded can on the same trial divided by the activity score. Rats were allowed to drink freely for 1 h at the end of the experimental sessions.

#### 2.7.1. Experimental Design

##### Shaping Period

During this 2-day session, rats started familiarising themselves with the arena. On the first day, rats were presented with seven white cans (modified from [54]), whose bottom was filled with 0.3 mL tap water. Rats had 20 min to explore the arena and take water from the cans. Animals were then removed and placed in their home cages. On the second day, only three cans randomly selected were rewarded with water, and rats were allowed to drink water for 10 min. After a 15-s interval, the procedure was repeated.

##### Spatial Task

Twenty-four h after the end of the shaping period, on three consecutive days and along ten trials per day, rats were placed in the same arena as the shaping period. The single rewarded can was placed in a fixed position among the empty non-rewarded cans. Rats could spend up to 180 s per trial visiting the cans and drinking water; once the reward was obtained, the rat was instantly removed from the arena. During the 15-s interval between trials, rats were placed in a small Plexiglas box (50 cm × 30 cm × 30 cm).

##### Simple Visual Task

Forty-eight h after the end of the spatial task, rats were placed in the arena where the rewarded can had a different appearance (i.e., a Pepsi can) than the other identical six cans and was randomly located on each trial. As in the previous step, rats could spend up to 180 s per trial visiting the cans and taking water.

### 2.8. Barnes Maze Test

To assess spatial memory in an aversive context, the adolescent male rat offspring underwent the Barnes maze test [52]. The apparatus consisted of a circular, grey platform made in Plexiglas with a diameter of 122 cm and a height of 90 cm. Twenty holes with a diameter of 10 cm were placed on the perimeter of the platform, and only one hole, namely the target hole, led to an under-platform chamber with dimensions of 12 cm × 12 cm × 35 cm—the escape box. The other holes were covered underneath with a flat box and looked identical to the other. In the task, the rat was placed in the middle of the platform and was initially unable to pinpoint the escape box, whose location varied according to the task phase. An additional aversive incentive was provided during the task in the form of bright lighting—two light spots placed 1.5 m above the platform with a power of 500 W each. Intense light and open supra-elevated spaces are aversive to rats and thus serve as motivating factors to induce escape behaviour [52]. Indeed, rodents find open well-lit spaces aversive, searching around the platform to find the escape box [59]. Additionally, on the walls of the laboratory room, different visual hints were provided in the form of large colourful geometric figures to facilitate the animal’s spatial orientation. The Barnes maze test consisted of the following phases: habituation; acquisition phase; probe task; reversal task [60]. The experimental design was developed as follows.

#### 2.8.1. Experimental Design

##### Habituation

Twenty-four hours before the acquisition phase, the rats were habituated to the platform and the escape box. The animals were placed in the middle of the platform and allowed to explore the apparatus for 180 s freely.

##### Acquisition Phase

Twenty-four hours after habituation, the same rats started the acquisition phase. It included one training session per day for three consecutive days. Each training session consisted of three 180-s trials. The location of the escape box remained the same over all the acquisition trials. Rats were placed in the middle of the maze, covered with an opaque bucket. After a few seconds of delay, the bucket was lifted, so that the initial orientation of the animal varied randomly from trial to trial. The trial was completed after 180 s or when the animal entered the escape box. Immediately after the animal entered the escape box, the hole was covered for 30 s, and the light was switched off. If the animal did not enter the escape box within 180 s, it was gently oriented there by the experimenter. The entries in the escape box were recorded in each trial to assess spatial learning. The learning performance of the adolescent offspring was measured in terms of days elapsed to reach the learning criterion, i.e., the spontaneous entering of the escape box during the three daily trials over the acquisition phase.

##### Probe Task

Twenty-four hours later, during the probe task, the target hole was closed. The memory of the location of the escape box was assessed for 90 s. Primary latency—the time required for the rat to make initial contact with the target hole—was measured to assess memory retention.

##### Reversal Task

The reversal task was performed twenty-four hours after the probe task. At that time, the position of the escape box was rotated 180° to the original, and three 180-s trials were run in one day (modified from [60]). The latency to escape—latency to find the escape box—was recorded in each trial.

### 2.9. Tissue Collection and qRT-PCR Procedure

After the behavioural battery, a subset of adolescent male rats, randomly selected from the two experimental cohorts, was sacrificed, and the brains were rapidly removed. The hippocampus was rapidly dissected bilaterally, flash-frozen in dry ice, and stored at −80 °C until subsequent analysis. RNA was isolated using homogenisation in Trizol (Invitrogen) followed by chloroform layer separation and precipitation with isopropanol, plus 70% ethanol washes to remove any residual salts from the isopropanol RNA precipitation step [61]. RNA was resuspended with water and then quantified with NanoDrop (ND-1000 Spectrophotometer, Thermo Scientific, Wilmington, DE, USA). RNA was reverse-transcribed to cDNA (SuperScript IV Reverse Transcriptase, Invitrogen). cDNA was then diluted and mixed with PowerUp SYBR Green Master Mix (Applied Biosystems) and primers. Samples were then heated to 95 °C for 10 min, followed by 40 cycles of 95 °C for 15 s, 60 °C for 1 m, 95 °C for 15 s, 60 °C for 30 s, and 95 °C for 15 s. Gene expression analysis was performed using the delta–delta C(t) method. Primers employed are indicated in Table 1.

### 2.10. Statistical Analysis

When data exhibited normality and equal variance, the difference between groups was determined by employing either the unpaired parametric Student’s *t*-test or two-way analysis of variance (ANOVA) for repeated measures followed by a Bonferroni post hoc test, when necessary. Nonparametric tests were performed if data did not show normal distribution or equal variance. Rat performance to reach the learning criterion was analysed using Kaplan–Meier event analysis over the acquisition period in the Barnes maze test, and the resulting curves were compared by employing the log-rank Mantel–Cox test. Data are reported as mean ± SEM. Statistical analysis was performed using GraphPad Prism software (Version 9.4.1; GraphPad Software Inc.; San Diego CA, USA) and statistical significance was set at alpha = 0.05.

## 3. Results

### 3.1. Behavioural Assessment of the Offspring

#### 3.1.1. Prenatal THC Exposure Increases Locomotor Activity and Does Not Modify Exploration in the Adolescent Offspring

The offspring were tested in the open field arena to evaluate the effects of prenatal THC exposure on locomotor activity and exploration. The statistical analysis highlighted that prenatal THC exposure induced a significant increase in the TDT (U = 25, *p* = 0.0053; Figure 2a) of adolescent offspring compared to the CTRL group. No significant differences were observed in NCT (t = 0.8140, df = 21, *p* = 0.4248; Figure 2b).

#### 3.1.2. Prenatal THC Exposure Does Not Impair Emotional Reactivity in the Adolescent Offspring

Male adolescent rats were tested in the EPM to evaluate anxiety-like behaviour. Student’s *t*-test performed on data from the percentage of time spent on open arm/time on open and closed arms displayed no differences among the experimental groups (t = 0.5043, df = 23, *p* = 0.6188; Figure 3a), as shown by the percentage of open arm entries out of total entries (t = 0.8914, df = 22, *p* = 0.3824; Figure 3b). When the number of total entries was analysed to assess general activity, the *t*-test showed a significant difference between the two experimental groups, with the number of total entries of the pTHC group being higher than that of the control rats (t = 2.928, df = 22, *p* = 0.0078; Figure 3c).

#### 3.1.3. Prenatal THC Exposure Does Not Impact the Responsivity to Natural Reinforcing Stimuli

Adolescent male rats were subjected to the sucrose preference test to evaluate their response to a natural reward. Prenatal THC exposure did not affect sucrose preference, calculated as the percentage of the volume of the sucrose solution intake divided by the total fluid volume (water plus sucrose). Indeed, no significant difference between the experimental groups was found (t = 1.867, df = 7.099, *p* = 0.1035; Figure 4).

#### 3.1.4. Prenatal THC Exposure Impairs Spatial Memory and Object Recognition Memory of the Adolescent Offspring in a Reinforce-Motivated Can Test

The results of a two-way ANOVA for repeated measures on the correct responses of adolescent male rats showed a significant effect of days (F (2, 34) = 60.05, *p* < 0.0001), a main effect of pTHC (F (1, 17) = 12.42, *p* = 0.0026), and their interaction (F (2, 34) = 3.370, *p* = 0.0462). In detail, the post hoc analysis showed a decrease in correct responses in the adolescent pTHC rats compared to controls on days 2 (t = 3.084, df = 51.00, *p* = 0.0099) and 3 (t = 3.261, df = 51.00, *p* = 0.0059; Figure 5a). The main effect of days (F (2, 34) = 116.3, *p* < 0.0001), pTHC (F (1, 17) = 8.338, *p* = 0.0102), and their interaction (F (2, 34) = 19.13, *p* < 0.0001) were observed for the reference memory errors. In particular, the post hoc analysis showed an increase in reference memory errors in the adolescent pTHC rats compared to controls on day 2 (t = 6.117, df = 51.00, *p* < 0.0001; Figure 5b). No statistically significant difference was detected in working memory errors (days: F (2, 34) = 5.562, *p* = 0.0081; pTHC: F (1, 17) = 0.7443, *p* = 0.4003; days × pTHC: F (2, 34) = 2.5001, *p* = 0.1442; Figure 5c) and activity score (days: F (2, 34) = 0.5443, *p* = 0.5852; pTHC: F (1, 17) = 0.02419, *p* = 0.8782; days × pTHC: F (2, 34) = 3.048, *p* = 0.0606; Figure 5d).

The offspring’s ability to discriminate a different object among identical ones, independently of its position, was assessed in the simple visual task, where pTHC rats displayed a significant deficit. Indeed, the results of a two-way ANOVA for repeated measures showed a main effect of days (F (2, 34) = 21.22, *p* < 0.0001) and a decreasing effect of prenatal THC exposure (F (1, 17) = 14.75, *p* = 0.0013), but not their interaction (F (2, 34) = 2.307; *p* = 0.1149) (Figure 6a), for correct responses. For reference memory errors, the effect of days (F (2, 34) = 18.94, *p* < 0.0001) and the increasing effect of pTHC (F (1, 17) = 21.83, *p* = 0.0002) were highlighted, but not their interaction (F (2, 34) = 1.814, *p* = 0.1784; Figure 6b). No statistically significant effect of prenatal THC exposure was observed in working memory errors (days: F (2, 34) = 13.39, *p* < 0.0001; pTHC: F (1, 17) = 0.1862, *p* = 0.6715; days × pTHC: F (2, 34) = 0.06036, *p* = 0.9415; Figure 6c) and activity score (days: F (2, 34) = 0.000, *p* > 0.9999; pTHC: F (1, 17) = 0.000, *p* > 0.9999; days × pTHC: F (2, 34) = 0.000, *p* > 0.9999; Figure 6d).

#### 3.1.5. Prenatal THC Exposure Impairs Spatial Memory and Reversal Learning in an Aversive Context

Spatial memory in an aversive context was assessed by employing the Barnes maze test, with 2-way ANOVA revealing a main effect of days (F (1.849, 94.28) = 78.43, *p* < 0.0001) and of pTHC (F (1, 51) = 4.878, *p* = 0.0317), but not their interaction, on latency to reach the escape box along the acquisition phase of adolescent male rat offspring. The performance of the adolescent offspring was measured in terms of days elapsed to achieve the learning criterion along the acquisition phase; Kaplan–Meier analysis showed that the mean learning days were 1.458 (95% Confidence Interval: 1.129–1.787) for CTRL rats and 1.759 days (95% Confidence Interval: 1.427–2.090) for THC-treated prenatal rats. The Log-rank (Mantel–Cox) test indicated no significant differences in learning performance (χ^2^ = 2.608, df = 1, *p* = 0.1064).

To assess spatial memory retrieval, pTHC-exposed adolescent rats were evaluated in the probe task of the Barnes maze. The results of the Student’s *t*-test on primary latency, which is the time required for the rat to make initial contact with the target hole, showed a significant increase in pTHC rats when compared to CTRL offspring (t = 2.324, df = 22, *p* = 0.0297; Figure 7a). Further, the effect of prenatal exposure to THC on reversal learning was assessed in the reversal task. pTHC exposure produced a significant increase in the latency to escape (t = 2.921, df = 22, *p* = 0.0079; Figure 7b) when the position of the escape box was rotated 180°.

### 3.2. Prenatal THC Exposure Modifies Markers of Synaptic Plasticity in the Hippocampus of Adolescent Offspring

To assess the effect of prenatal exposure to THC on the hippocampal markers of neuroplasticity in the adolescent male rat offspring, we examined the mRNA relative expression levels of the main effectors of excitatory transmission and ECS-driven downstream signalling that regulate synaptic strength.

Data from the Student’s *t*-test showed a significant effect of prenatal THC exposure in increasing the relative expression levels of the obligatory NMDAR NR1 subunit, which controls the magnitude of the excitatory signalling, in the hippocampus of the adolescent offspring (t = 17.39, df = 8, *p* < 0.0001; Figure 8a). On the other hand, the levels of the NR2A subunit, as the synaptic player crucial for the functionality of the excitatory signalling, and PSD-95 levels, the scaffolding protein critical for the PSD localisation of NR2A, appeared to be decreased by pTHC exposure (NMDAR NR2A subunit: t = 2.859, df = 8, *p* = 0.0212, Figure 8b; PSD95: U = 2, *p* = 0.0333, Figure 8c). Moreover, our data showed that pTHC exposure induced an increase in the expression of the molecular effectors that prompt ECS-mediated long-term control of the synaptic strength, i.e. mGluR5 (U = 0, *p* = 0.0095; Figure 8d) and its scaffolding protein Homer1 (U = 0.00, *p* = 0.0095; Figure 8e). In addition, pTHC adolescent progeny displayed increased levels of CB1R (U = 0, *p* = 0.0095; Figure 8f), together with increased expression of its functional effector HINT1 (t = 2.789, df = 8, *p* = 0.0236; Figure 8g).

## 4. Discussion

This research, through the use of a valid rodent model of gestational exposure to mild THC concentration [48,62], provides evidence that prenatal THC exposure is associated with deficits in spatial memory formation, object recognition memory and reversal learning in the adolescent male rat offspring. The behavioural outcomes are paralleled by molecular abnormalities that resemble abnormal gene expression of glutamatergic- and CB1R-related markers in the hippocampus, pointing overall to pTHC as a complicating player of physiological neurodevelopment. This and other research groups have shown that the detrimental effect of prenatal exposure to cannabinoids on cognitive processes appeared when the animals were challenged in an emotionally relevant cognitive task [44,63,64,65]. Thus, here we aimed at studying spatial memory in two motivated memory tasks exploiting a reward- or a fear-driven cognitive response.

Spatial memory formation and recognition memory were first assessed in the Can test, a reinforce-motivated memory task in which thirsty rats were trained to identify a single water-rewarded can among a set of seven, based on its position or its different appearance [54]. The Can test exploits the rat’s ability to create a map of the surrounding environment, which occurs trial by trial and day by day [66]. The ability to acquire and retrieve the position of objects in space and to associate them with different stimuli is crucial for survival. It is dynamically processed in the hippocampus, i.e., the core of the neural memory system that provides a spatial framework to the memory traces [67,68]. Our findings show that prenatal THC exposure did not affect rat learning performance on the first day of the Can test, while there were decreased correct responses and increased reference memory errors on the second and third days. This suggests that prenatal THC exposure, rather than affecting the acquisition of the spatial information helpful to the formation of the cognitive map, seems to impair the temporary consolidation and retrieval of the spatial memory traces in a reinforce-motivated context. This result is not unexpected because THC—as well as synthetic cannabinoids—impairs memory formation and consolidation in emotionally salient contexts by the activation of CB1R, confirming a selective role of hippocampal CB1R-related signalling in consolidation, but not acquisition, of new memories [69,70,71]. This evidence is consistent with human data showing that cannabinoids impair the consolidation and retrieval of both verbal and nonverbal information [72,73,74] and reversal learning [75].

The second task of the Can test was aimed at assessing object recognition memory, by investigating rats’ ability to recognise an unlike can whose position changed among a set of identical ones [54]. Our results indicate that prenatally THC-exposed rats display an impairment in memory formation when they are requested to remember the unlike rewarded can, independently from its position, day by day. These results are consistent with data from the adolescent offspring of mothers exposed to higher doses of THC that exhibited deficits in visual discrimination in the novel object recognition test (NORT) [76] and alterations in main hippocampal oscillations [63]. Indeed, if the perirhinal and entorhinal cortices take part in object recognition memory, and particularly in object discrimination when complex stimuli must be processed [77,78,79], the hippocampus operates as the hub where direct and indirect projections carrying spatial/object memory information are integrated and encoded [80,81,82].

The Can test takes advantage of normal motor activity and motivation to visit the cans (thirst) to accelerate the achievement of the task. Accordingly, one might speculate that, rather than specifically affecting cognitive processing, prenatal THC exposure could be associated with a low incentive drive to visit the cans, a weak sensitivity for the natural stimulus, or depressed locomotor activity. Indeed, rats were sufficiently motivated to visit at least one can in eight or nine out of the ten trials; plus, the number of visits recorded throughout the Can test was not different between the experimental groups. THC did not modify sucrose preference in adolescent rats, but increased locomotor activity while not modifying exploration, as shown in the open field test. Moreover, we observed no impairment in working memory, a complex cognitive function in which sustained attention plays a substantial role [83]. Overall, this evidence suggests that rather than inducing attentional deficits, the effect of prenatal THC, in our experimental conditions, involves specific microcircuits for spatial- and object-recognition reference memory in the adolescent male offspring.

To extend our assessment of THC effects into an aversive emotionally-salient context [84,85,86], rats were tested in the Barnes maze. It is a rodent dry-land-based, hippocampus-dependent behavioural paradigm which allows the evaluation of the rat’s attentional resources towards the goal of safety [52,87,88]. Indeed, bright light and open spaces are aversive to rodents, which are motivated to escape by searching around the platform and finding a safe shelter in the escape box [59]. Prenatally THC-exposed adolescent rats did not display deficits in spatial acquisition in the Barnes maze, whereas they displayed impairment in remembering the escape box’s location and detecting its new position. This suggests that in-utero exposure to THC was associated with an impairing effect on hippocampus-related memory processing and neural circuits processing reversal learning. The Barnes maze implies that rats learn to solve a maze by using environmental cues in an anxiogenic context. Therefore, the performance of rodents in the Barnes maze may be influenced by non-cognitive factors, such as the emotional state [52]. However, EPM data showed no difference in anxiety-like behaviour between the experimental groups, ruling out a major interference of emotional dysregulation in prenatally THC-exposed rats. Thus, our data suggest that the impaired cognitive performance of the adolescent rats prenatally exposed to THC may reflect a complex, specific interaction between prenatal THC and the neurodevelopmental trajectories underlying the formation of spatial- and object-recognition memory.

Cannabinoid-induced cognitive deficits are known to be associated with alterations of hippocampal glutamatergic synaptic plasticity at multiple levels, thus recapitulating findings from earlier neurophysiology studies linking altered CB1R function to abnormal hippocampal long-term potentiation, memory encoding, and glutamate release [65,89,90]. In detail, we observed a dramatic increase in NR1 and a concomitant decrease in NR2A subunits and PSD95 mRNA expression in the hippocampus of the adolescent male rat offspring. Therefore, our results suggest that prenatal exposure to THC promoted a complex rearrangement in the relevant players of the excitatory synapse, i.e., NMDAR, PSD95, and mGluR5/Homer1 complex, which is also associated with an abnormal mRNA expression of CB1R.

The majority of hippocampal NMDAR subunits display the association of NR1–NR2 complexes at the synaptic level [91,92], with specific subunits arrangements reflecting not only the receptor functional properties but also the strength of the excitatory synapse [93,94,95]. The NR1 subunit is required for NMDA receptor activation [96,97], for the regulation of the channel’s properties, directly influencing the strength of the excitatory signalling [26,27,91]. On the other hand, the NR2A subunit is mandatory for the NMDAR efficient binding to PSD-95; this interaction favours NMDAR clustering and stabilisation on the postsynaptic membrane and contributes to dynamic changes in the synapse [30,31]. The opposite changes observed in the NR1 and NR2A subunit mRNA in the hippocampus of in-utero THC-exposed rats suggest distinct regulatory mechanisms that may contribute to alterations in NMDAR levels or the proportion of receptor populations with different subunit combinations [98,99]. Because many of the physiological and pharmacological properties of NMDAR depend on the recruitment of specific subunits [100] and the functional support of PSD-95, our evidence suggests that prenatal exposure to THC can promote the occurrence of an abnormal NMDAR/PSD-95 combination as a feature of a dysfunctional glutamatergic synapse.

Intriguingly, we also measured an increase in the expression of mGluR5-and Homer1 mRNA in the hippocampus of the adolescent offspring prenatally exposed to THC. Indeed, by coupling with Homer1 proteins, mGluR5 stimulates eCBs-driven downstream signalling that elicits long-lasting depression of excitatory synaptic transmission, i.e., mGluR5-LTD [14,21]. This suggests that the functional consequences of pTHC exposure may also include augmentation in the effectors of eCBs-induced LTD, likely in response to a disruption in the gain and loss of synaptic efficiency. Interestingly, we also measured a significant increase in CB1R mRNA in the hippocampus of the offspring prenatally exposed to THC. Accordingly, early studies report a significant increase in CB1R expression and mRNA levels in the hippocampus and the prefrontal cortex of perinatally exposed THC adult rats [101,102]. Although THC mode of action in the developing nervous system remains unknown, CB1R expressed by developing neurons emerges as a likely candidate [103,104]. Evidence indicates a role for CB1Rs in neuronal proliferation, specification, migration, and axon outgrowing [8,9], suggesting that the early manipulation of endocannabinoid signal can hijack the physiological wiring of neuronal networks. Indeed, in-utero exposure to CB1R agonists significantly curtails the strength of GABAergic circuits by reducing the density of CB1R-containing inhibitory (CCK^+^) interneurones (CCK^+^-IN) [63]. These anatomical defects result in deficits in CCK^+^-IN mediated feedforward/feedback inhibition in the CA1 microcircuit, excitatory/inhibitory imbalance in specific neuronal circuits, and, lastly, altered cognitive functions [64]. Given our data, we cannot prove a mechanistic explanation of the behavioural and molecular changed we report here; nevertheless, it is tempting to speculate that as a consequence of pTHC exposure, a substantial reduction in feedback inhibition is produced by a functional antagonism of hippocampal CB1R signalling [105]. The increase in CB1R mRNA observed in our experimental conditions might, thus, provide an adaptive mechanism that scales specific CB1R^+^ synapses and balance the putative asymmetry between inhibitory and excitatory tone. Intriguingly, in line with the pTHC-induced increase in the expression of NR1 subunit and CB1R, here we report an augmentation in the mRNA expression of HINT1, a scaffold protein that, by the direct link of the CB1R C-terminus to the NR1 subunit, plays a crucial role in preventing NMDAR over-activation [106]. In detail, the complex CB1R-HINT1-NR1 is co-internalized, whereas the HINT1 protein conserves its association with the CB1R and is recycled back to the cell surface; the NR1 subunit is released in the cytoplasm, facilitating the disassembling of the NMDAR [33]. Under the proposed interpretation, the increase in HINT1 mRNA expression observed in this study could serve to enable CB1R-NMDAR cross-regulation in the context of excessive NR1 expression. Therefore, the overexpression of mGluR5/Homer1 signalling and CB1R/HINT1 cooperation might be functionally interpreted as an attempt to offset the excitatory-inhibitory imbalance and mitigate the dysfunctional consequences promoted by pTHC exposure.

However, if this sounds like a reasonable hypothesis, we may not rule out that other mechanistic dynamics can result from pTHC exposure. Indeed, as a consequence of the increased expression of both CB1R and HINT1 here observed, an augmented removal of the NR1 subunit from the cell membrane might result in glutamatergic hypo-function [33], thus promoting an increase in NR1 expression as a counter-adaptive mechanism. Future research is needed to evaluate dynamic changes in neuroplasticity as a consequence of environmental challenges in pTHC-exposed offspring and confirm, as the literature supports [107], that differences in mRNA expression reflect changes in the respective protein levels.

## 5. Conclusions

Overall, here we provide evidence of pTHC exposure -related long-lasting consequences in adolescent male rat offspring, which include deficits in spatial memory formation, object recognition memory, reversal learning, in both appetitively- and aversively-motivated conditions and modifications in the expression of receptors and scaffolding partners essential for the efficiency and magnitude of the hippocampal excitatory synapse. We hypothesise the occurrence of an altered neurodevelopmental trajectory associated with prenatal exposure to THC that leads to a specific impairment in spatial cognitive processing in the offspring through a perturbation in the effectors of eCBs-related excitatory/inhibitory balance in specific hippocampal neural circuits. This research strongly prompts increased awareness of the memory-disruptive effects of prenatal exposure to THC and highlights novel dysfunctional hippocampal players that may represent future pharmacological targets.

## Figures and Tables

**Figure 1 pharmaceutics-15-00692-f001:**
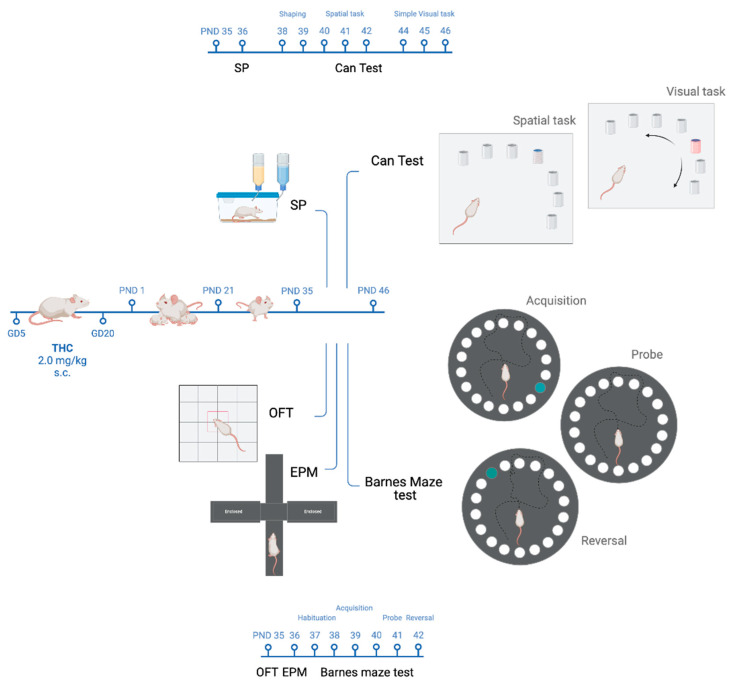
Behavioural procedures. Adolescent male rat offspring prenatally exposed to either vehicle (CTRL) or THC (2 mg/kg; pTHC) underwent behavioural assessments starting from PND 35 to 46. GD, gestational day; PND, postnatal day; s.c., subcutaneous; THC, Δ9-tetrahydrocannabinol; OFT, open field test, EPM, elevated plus maze test; SP, sucrose preference test. Created with BioRender.com, https://app.biorender.com (accessed on 23 January 2023).

**Figure 2 pharmaceutics-15-00692-f002:**
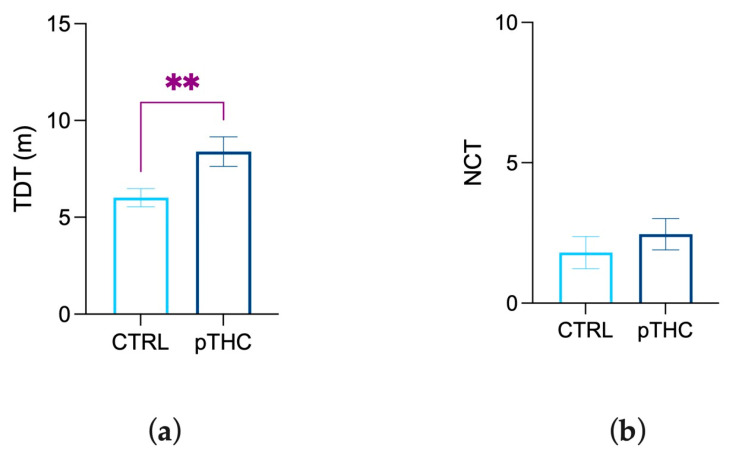
Effects of prenatal TCH exposure (2 mg/kg) on locomotor activity and exploration of the adolescent male rat offspring in the OFT: (**a**) Prenatal exposure to THC increased locomotor activity in terms of TDT while (**b**) induced no significant effects in exploratory activity in terms of NCT. Each bar represents the mean of n = 12 rats; error bars indicate SEM. ** *p* < 0.01; CTRL = male rat offspring prenatally exposed to Veh; pTHC = male rat offspring prenatally exposed to THC; TDT = total distance travelled; NCT = the number of central transitions from the peripheral to the central area of the arena.

**Figure 3 pharmaceutics-15-00692-f003:**
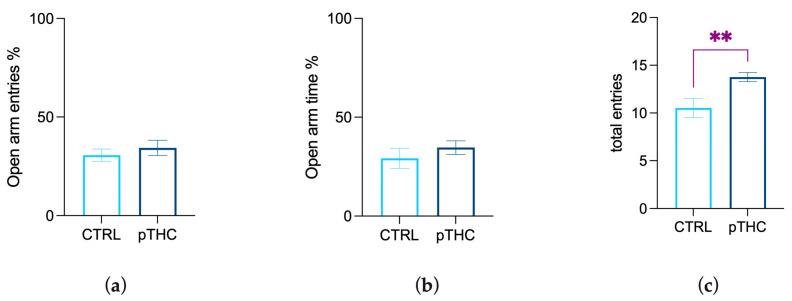
Effects of prenatal TCH exposure (2 mg/kg) on emotional reactivity of the adolescent male rat offspring in the EPM: (**a**) Prenatal exposure to THC did not affect the percentage of time spent on the open arm and (**b**) on the percentage of entries in the open arm, while (**c**) increasing the number of total entries. Each bar represents the mean of n = 12 rats; error bars indicate SEM. ** *p* < 0.01; CTRL = male rat offspring prenatally exposed to Veh; pTHC = male rat offspring prenatally exposed to THC; Open arm time % = the percentage of time spent on the open arm/total time spent on open and closed arms; Open arm entries % = the percentage of the number of entries in the open arm/number of entries in the open and closed arms.

**Figure 4 pharmaceutics-15-00692-f004:**
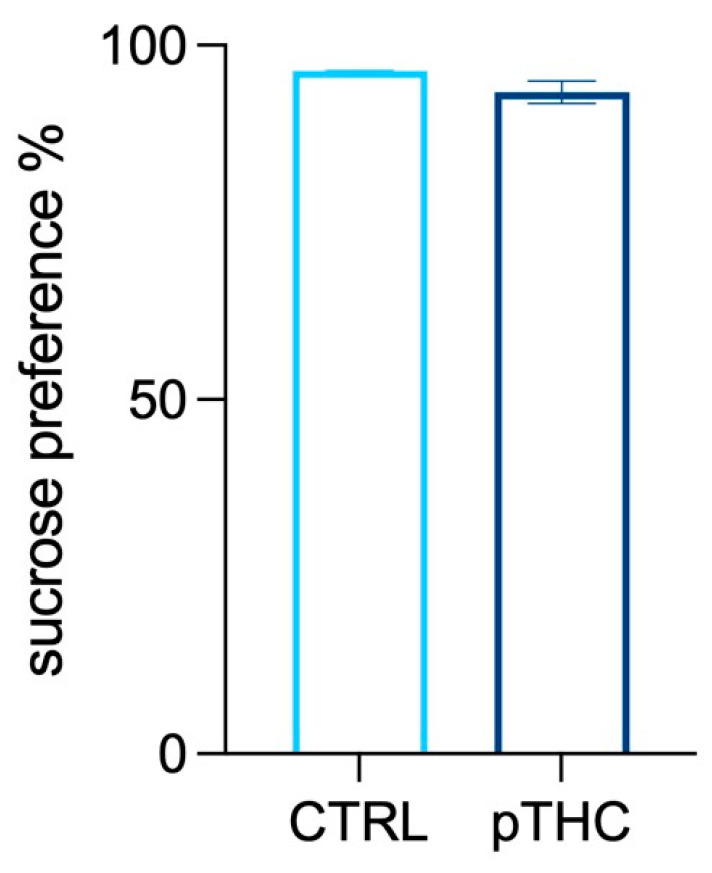
Effects of prenatal TCH exposure (2 mg/kg) on the responsivity to natural reinforcing stimuli of the adolescent male rat offspring in the sucrose preference test. Prenatal exposure to THC did not affect sucrose preference, calculated as the percentage of the volume of the sucrose solution intake divided by the total fluid volume (water plus sucrose). Each bar represents the mean of n = 9 rats in the CTRL group and n = 8 rats in the pTHC group; error bars indicate SEM. CTRL = male rat offspring prenatally exposed to Veh; pTHC = male rat offspring prenatally exposed to THC.

**Figure 5 pharmaceutics-15-00692-f005:**
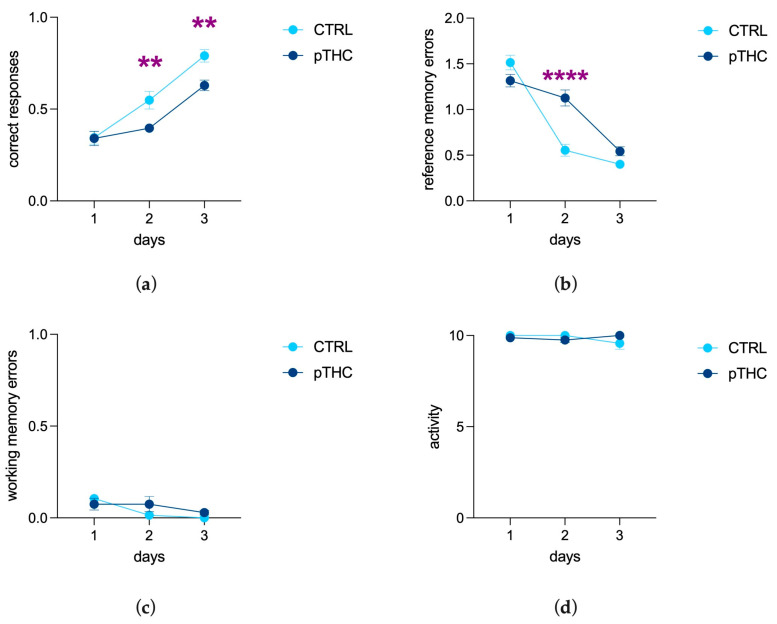
Effect of prenatal THC exposure (2 mg/kg) on spatial memory of the adolescent male rat offspring in the spatial task of the Can test: (**a**) pTHC decreased the correct responses of adolescent male rats on days 2 and 3 and (**b**) induced an increase in reference memory errors on day 2 of adolescent male rats compared to controls. (**c**) No effect of prenatal THC treatment was detected in working memory errors and (**d**) activity. Each bar represents the mean of n = 9 rats in the CTRL group and n = 10 rats in the pTHC group; error bars indicate SEM. ** *p* < 0.01; **** *p* < 0.0001; CTRL = male rat offspring prenatally exposed to Veh; pTHC = male rat offspring prenatally exposed to THC.

**Figure 6 pharmaceutics-15-00692-f006:**
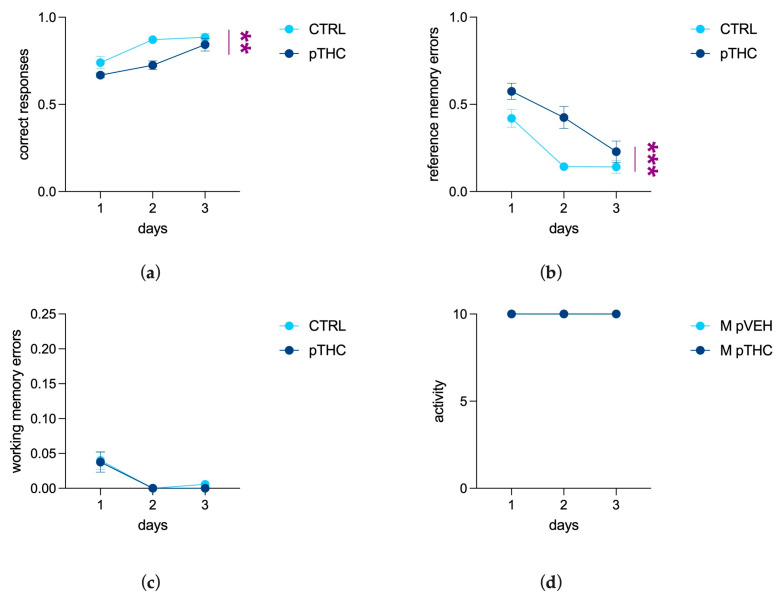
Effect of prenatal THC exposure (2 mg/kg) on the object recognition memory of the adolescent male rat offspring in the simple visual task of the Can test: (**a**) Prenatal exposure to THC decreased correct responses and (**b**) increased reference error memory of adolescent male rats. (**c**) No significant effect of prenatal THC treatment was detected in working memory errors and (**d**) activity. Each bar represents the mean of n = 9 rats in the CTRL group and n = 10 rats in the pTHC group; error bars indicate SEM. ** *p* < 0.01; *** *p* < 0.001; CTRL = male rat offspring prenatally exposed to Veh; pTHC = male rat offspring prenatally exposed to THC.

**Figure 7 pharmaceutics-15-00692-f007:**
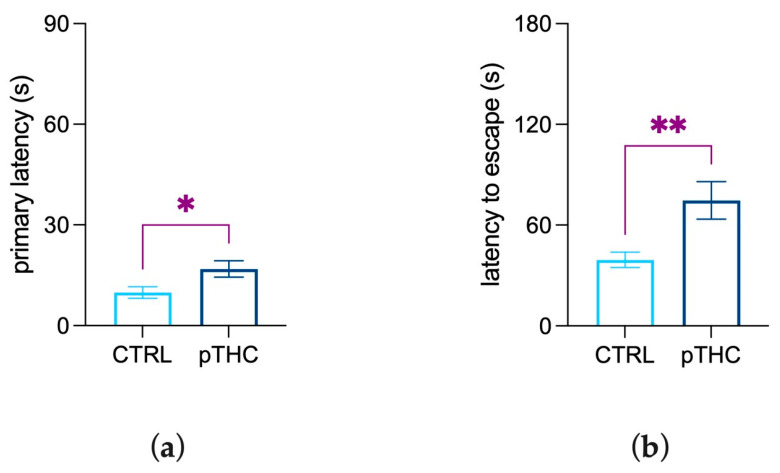
Effect of prenatal THC exposure (2 mg/kg) on the spatial memory and reversal learning of the adolescent male rat offspring in the Barnes maze test. pTHC increased: (**a**) primary latency and (**b**) latency to escape in the probe and reversal tasks, respectively. Each bar represents the mean of n = 12 rats; error bars indicate SEM. * *p* < 0.05; ** *p* < 0.01; CTRL = male rat offspring prenatally exposed to Veh; pTHC = male rat offspring prenatally exposed to THC.

**Figure 8 pharmaceutics-15-00692-f008:**
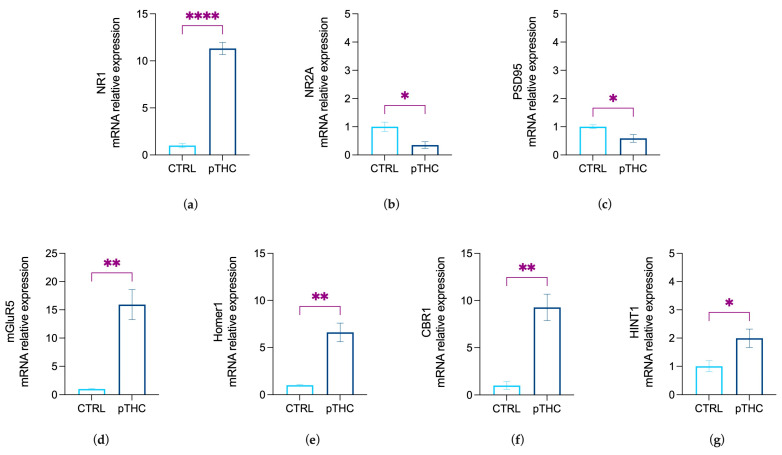
Prenatal THC exposure (2 mg/kg) alters the mRNA relative expression levels of the effectors of excitatory and ECS-driven downstream signalling in adolescent rats. Prenatal THC exposure (**a**) increased the expression levels of the NMDAR NR1 subunit and decreased (**b**) the levels of NR2A and (**c**) PSD95 in the hippocampus of the adolescent male rat offspring. (**d**) pTHC induced an increase in mGluR5, as well as (**e**) its scaffolding partner Homer1 isoform in the hippocampus of the adolescent progeny. Lastly, (**f**) CB1R and (**g**) HINT1 were increased by pTHC in the hippocampus of the adolescent male offspring. Data are shown as the mean ± SEM. * *p* < 0.05; ** *p* < 0.01; **** *p* < 0.0001; CTRL = male rat offspring prenatally exposed to Veh; pTHC = male rat offspring prenatally exposed to THC.

**Table 1 pharmaceutics-15-00692-t001:** Primers employed in qRT-PCR experiments.

Gene Name	Primer Sequence	Product
Gapdh	GTTTGTGATGGGTGTGAACC (Forward) CTTCTGAGTGGCAGTGATG (Reverse)	
NMDAR NR1 subunit-Grin1		Rn_Grin1_1_SG QuantiTect Primer Assay (QT00182287)
NMDAR NR2 subunit-Grin2A		Rn_Grin2a_1_SG QuantiTect Primer Assay (QT00379281)
PSD-95-Dlg4		Rn_Dlg4_1_SG QuantiTect Primer Assay (QT00183414)
mGluR5-Grm5		Rn_Grm5_1_SG QuantiTect Primer Assay (QT01081549)
Homer1-HOM1	CTTCACAGGAATCAGCAGGAG (Forward) GTCCCATTGATACTTTCTGGTG (Reverse)	
CB1R-Cnr1		Rn_Cnr1_1_SG QuantiTect Primer Assay (QT00191737)
Histidine triad nucleotide-binding protein 1 (HINT1)		Rn_Hint1_1_SG QuantiTect Primer Assay (QT01602713)

## Data Availability

The data presented in this study are available on request from the corresponding author.

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
