# Peer review of "Prenatal Exposure to Δ9-Tetrahydrocannabinol Affects Hippocampus-Related Cognitive Functions in the Adolescent Rat Offspring: Focus on Specific Markers of Neuroplasticity"

_pharmaceutics, 2023, doi:10.3390/pharmaceutics15020692_

Round 1

Reviewer 1 Report

The study of Castelli and colleagues sets out to determine whether prenatal exposure to the cannabis component THC leads to identifiable cognitive and neuroplastic deficits in the male offspring. The focus is put on the hippocampus and, accordingly, different spatial and emotional metrics are obtained. Accompanying these behavioral findings, the Authors determine the genetic expression of different mRNAs that encode proteins with major neuroplastic roles within the hippocampal circuitry. With the exemption of the lack of female subjects, the behavioral findings are sound, and the experiments seem well designed and performed. The conclusions about the pTHC-induced cognitive deficits are well justified. My problem with their work comes with the conceptual and experimental treatment given to the neuroplastic consequences of pTHC. More specific comments on this below.

· “Neuroplasticity”: Any given plastic phenomena is framed within a dynamic framework. If you measure something only once, you cannot ascertain whether it reacted differently to a stimulus (as we consider plasticity roughly is), or it simply was baseline different from the very beginning. A single-point change in the expression of a given mRNA can hardly inform about any kind of neuroplastic process. To be able to talk about plasticity, Authors should have extracted brain samples at different timepoints (from different subjects, given the ex vivo approach followed) to track how the mRNA’s expression adapts to the stimuli of choice (it may be the behavioral testing itself, a stressor, different stages of development…). A plethora of ex vivo electrophysiological experiments can also be performed to answer these questions at different timescales. However, as it is shown here, it is not possible to determine whether the expression of the mRNAs of interest is baseline different or whether it changes in reaction to a given neuroplastic-triggering event. A second critique to the Authors’ articulation of the “neuroplasticity” concept is related to the conceptual distances existing between mRNA, protein expression and function. It is possible that mRNA expression is different but protein expression of NR1-2A subunits, mGluR5, HINT1, CB1R, PSD95 and Homer1 remains similar between groups. I would suggest the Authors either change the wording of the Manuscript away from the concept of plasticity (of course it can be suggested in the Discussion) or perform additional experiments to better delineate the putative neuroplastic changes. Performing mRNA and protein measurements at different time points, in relation to an event that is hypothesized to trigger hippocampal neuroplasticity, would be required for this second option.  

· In my humble, non-native speaker opinion, the Manuscript would greatly benefit from an English editing service. The introduction and discussion sections feel considerably wordy, too long, and some sentences are not readable. I had even greater problems trying to read two introduction sentences going from lines 42 to 46 and 53 to 61.

Author Response

The study of Castelli and colleagues sets out to determine whether prenatal exposure to the cannabis component THC leads to identifiable cognitive and neuroplastic deficits in the male offspring. The focus is put on the hippocampus and, accordingly, different spatial and emotional metrics are obtained. Accompanying these behavioral findings, the Authors determine the genetic expression of different mRNAs that encode proteins with major neuroplastic roles within the hippocampal circuitry. With the exemption of the lack of female subjects, the behavioral findings are sound, and the experiments seem well designed and performed. The conclusions about the pTHC-induced cognitive deficits are well justified. My problem with their work comes with the conceptual and experimental treatment given to the neuroplastic consequences of pTHC. More specific comments on this below.

“Neuroplasticity”: Any given plastic phenomena is framed within a dynamic framework. If you measure something only once, you cannot ascertain whether it reacted differently to a stimulus (as we consider plasticity roughly is), or it simply was baseline different from the very beginning. A single-point change in the expression of a given mRNA can hardly inform about any kind of neuroplastic process. To be able to talk about plasticity, Authors should have extracted brain samples at different timepoints (from different subjects, given the ex vivo approach followed) to track how the mRNA’s expression adapts to the stimuli of choice (it may be the behavioral testing itself, a stressor, different stages of development…). A plethora of ex vivo electrophysiological experiments can also be performed to answer these questions at different timescales. However, as it is shown here, it is not possible to determine whether the expression of the mRNAs of interest is baseline different or whether it changes in reaction to a given neuroplastic-triggering event.

A second critique to the Authors’ articulation of the “neuroplasticity” concept is related to the conceptual distances existing between mRNA, protein expression and function. It is possible that mRNA expression is different but protein expression of NR1-2A subunits, mGluR5, HINT1, CB1R, PSD95 and Homer1 remains similar between groups. I would suggest the Authors either change the wording of the Manuscript away from the concept of plasticity (of course it can be suggested in the Discussion) or perform additional experiments to better delineate the putative neuroplastic changes. Performing mRNA and protein measurements at different time points, in relation to an event that is hypothesized to trigger hippocampal neuroplasticity, would be required for this second option.  

We thank the Reviewer for the interesting standpoint which certainly represents a hint for future insights. 

In the current research, we cannot rule out that the alteration of the hippocampal markers occurs by the interaction between the gestational background and the behavioural tasks. As a matter of fact, both groups are exposed to the same tasks; what we highlight is that the prenatal THC exposure is able to modify the response of the hippocampal glutamatergic- and eCB- related markers to the environmental stimuli we challenge the animal with.

The lack of protein measurement is certainly a limitation of our study, as indicated in the text (lines 608-613). We cannot provide evidence of the parallel increase in the genic expression of the markers and their relative protein expression. However, in light of the impaired cognitive performance observed, it is likely that the protein expression of the markers under investigation is affected too by prenatal THC exposure. Those analyses will be performed soon in our lab and will be the subject of future publications.

In my humble, non-native speaker opinion, the Manuscript would greatly benefit from an English editing service. The introduction and discussion sections feel considerably wordy, too long, and some sentences are not readable. I had even greater problems trying to read two introduction sentences going from lines 42 to 46 and 53 to 61.

We thank the Reviewer for contributing to the improvement of the manuscript comprehension. The sentences have been properly modified in the text.

Reviewer 2 Report

The manuscript provides data about long-lasting effects of prenatal THC-exposure in adolescent male rat offspring. Deficits in spatial memory formation, object recognition memory, and reversal learning both in rewarding and aversively motivating conditions were found. Modified expression at mRNA level of receptors and scaffolding proteins, essential for the efficiency and magnitude of the hippocampal excitatory synapse, was also shown.

 The results are valuable, add some knowledge to understanding the effect of prenatal THC-exposure on neurodevelopment and its consequences in adolescent using animal model. The altered mRNA expression would be nice to confirm by protein expression data.

 Minor comments

-        Fig. 2-7: give the number of animals in the Veh and pTHC groups used in the experiments.

-        What was the total number of male offspring used in the study?

-        The number of references needs reduction. Please be careful about the proper reference is indicated in the text.

-          In Section 2.9 use min instead of m as abbreviation of minutes.

-          Table 1: the title should be above the table.

Author Response

The manuscript provides data about long-lasting effects of prenatal THC-exposure in adolescent male rat offspring. Deficits in spatial memory formation, object recognition memory, and reversal learning both in rewarding and aversively motivating conditions were found. Modified expression at mRNA level of receptors and scaffolding proteins, essential for the efficiency and magnitude of the hippocampal excitatory synapse, was also shown.

 The results are valuable, add some knowledge to understanding the effect of prenatal THC-exposure on neurodevelopment and its consequences in adolescent using animal model. The altered mRNA expression would be nice to confirm by protein expression data.

We are thankful to the Reviewer for the comments and we appreciate the suggestion that is, actually, the on-going work of this research group.

 Minor comments

-        Fig. 2-7: give the number of animals in the Veh and pTHC groups used in the experiments.

We are grateful to the Reviewer for the suggestion. The number of animals in the experimental groups has been added in the text (line 134).

-        What was the total number of male offspring used in the study?

The total number of male offspring used in the study has been added in section “2.1. Animals and treatment”.

-        The number of references needs reduction. Please be careful about the proper reference is indicated in the text.

We appreciate the Reviewer’s comment, proper modifications to the reference list have been made.

-          In Section 2.9 use min instead of m as abbreviation of minutes.

We thank the Reviewer for the suggestion. The modification has been made in the text.

-          Table 1: the title should be above the table.

We thank the Reviewer for the comment. The correction has been made in the text.

Reviewer 3 Report

Generally: extreme long and complicated sentences – especially in the introduction.

49        In force of the large distribution – is not correct, please re-phrase

53-61 Write in separate sentences.

64        subunits edition – I do not understand what does it mean

68-70 Re-phrase the sentence, it is hard to understand in the present form

75        Here the authors should add how activation of mGluR5 leads to the generation of the endocannabinoid 2-arachidonoylglycerol

79-81 The fact the ECS is present in the developing brain and involved in brain development (as written in 46-49) is the reason that prenatal THX exposure has negative effects and not what the authors wrote.

81        Please separate clearly prenatal and adolescent exposure – this paper deals only with the first.

135      Time between the behavioural tests is missing. Could be shown as a time-line similarly as in the nice Figure 1.

156      How were the animals tracked?

200      What is the activity score?

215      On centered? What does it mean?

234      Two points of light?

258      Learning criterion was rather low: animals can find accidentally the escape whole.

294      The term “ behavioural reactivity” is very vague, it could be anything. Please be more specific. Moreover, here the take home message is a hyperactivity in the exposed rats, so we can´t say that there is no change.

341      Add to the title that in the Can test. Also please write instead of “alters” more specifically “impairs” in the title.

400      Delete “in the primary latency” because it is trivial from the first part of the sentence

411      “Neuroplasticity” is an unfortunate term here: no dynamic change in the reactivity of hippocampus was tested but the expression of selected genes.

443      The authors should add why they used only male rats in their study.

443      Previous results suggested a disturbance in emotional processing and memory after prenatal THC treatment. Present study found no change in anxiety, reward-related behaviour but in spatial and recognition memory. What could be the reason?

468      Although CB1R activation impairs memory formation, exposed mice were not treated with CB1 agonist before / during the test. What the authors found here is a long-term developmental consequence of THC exposure and not a direct effect of THC.

489      As a possible factor, which may contribute to the inferior performance of THC exposed rats attention deficit should be mentioned.

514      Previous studies found altered stress sensitivity (Lallai et al, 2022) and altered anxiety (Ellis et al, 2022) in exposed rats. The authors should discuss the reason of difference in their and previous findings.

518-527          The authors should compare their present findings in the hippocampus with their previous findings in the prefrontal cortex, amygdala and n.accumbens in the expression of target genes.

544      Using bulk expression analysis one can´t discriminate cell-line specific changes in the level of CB1 receptors, though CB1 activity on glutamatergic and GABAergic neurons has just an opposite effect on hippocampal activity. The authors should discuss this point.

Author Response

Generally: extreme long and complicated sentences – especially in the introduction.

49        In force of the large distribution – is not correct, please re-phrase 53-61 Write in separate sentences.

64        subunits edition – I do not understand what does it mean

68-70 Re-phrase the sentence, it is hard to understand in the present form

We thank the Reviewer for the suggestions. The changes have been produced in the text.

75        Here the authors should add how activation of mGluR5 leads to the generation of the endocannabinoid 2-arachidonoylglycerol

We thank the Reviewer for contributing to the improvement of the manuscript. The description has been added in the text.

79-81 The fact the ECS is present in the developing brain and involved in brain development (as written in 46-49) is the reason that prenatal THX exposure has negative effects and not what the authors wrote.

81        Please separate clearly prenatal and adolescent exposure – this paper deals only with the first.

We are grateful to the Reviewer for contributing to the improvement of the manuscript comprehension. The sentences have been properly modified in the text.

135      Time between the behavioural tests is missing. Could be shown as a time-line similarly as in the nice Figure 1.

Time between the behavioural tests have been added in figure 1.

156      How were the animals tracked?

As indicated in section 2.3, the behaviour of the rats was monitored by an experimenter and videos were recorder by an automatic video-tracking system, AnyMaze (Stoelting Europe, Dublin, Ireland). The behavior was then quantified by an experimenter who was not aware of the experimental groups.

200      What is the activity score?

The activity score is the number of trials on which rats visited at least one can (up to 10 during each experimental session), as stated in lines 211-212.

215      On centered? What does it mean?

234      Two points of light?

We are grateful to the Reviewer for contributing to the improvement of the manuscript comprehension. The sentences have been properly modified in the text.

258      Learning criterion was rather low: animals can find accidentally the escape whole.

We thank the Reviewer for raising the point. Actually, to exclude the fortuitous accomplishment of the task, we recorded the learning criterion when the animals found the escape box in all the three consecutive trials.

294      The term “ behavioural reactivity” is very vague, it could be anything. Please be more specific. Moreover, here the take home message is a hyperactivity in the exposed rats, so we can´t say that there is no change.

Thank you for noting that, it was an oversight. The parameter has been properly re-named.

341      Add to the title that in the Can test. Also please write instead of “alters” more specifically “impairs” in the title.

400      Delete “in the primary latency” because it is trivial from the first part of the sentence

The sentences have been properly modified in the text.

411      “Neuroplasticity” is an unfortunate term here: no dynamic change in the reactivity of hippocampus was tested but the expression of selected genes.

In the current research, we intended to assess the modification induced by prenatal THC on the plasticity of specific hippocampal glutamatergic- and eCB- related markers after the environmental stimuli we challenged the animal with.

443      The authors should add why they used only male rats in their study.

We are thankful to the Reviewer for raising the point. The current study is the first part of a larger research project that also involves the use of female rats. Off-the-record, the female offspring exposed to pTHC showed a similar impairment in the cognitive performance.

443      Previous results suggested a disturbance in emotional processing and memory after prenatal THC treatment. Present study found no change in anxiety, reward-related behaviour but in spatial and recognition memory. What could be the reason?

We thank the Reviewer for the comment. Our previous evidence (Brancato et al. J Psychopharmacol. 2020 Jun;34(6):663-679) showed no impact on emotional processing induced by gestational THC, yet an impairment in the formation of memory traces when integration between the environmental encoding and the emotional/motivational processing was required. Nonetheless, we reported that THC interference in the endocannabinoidergic signalling during pregnancy is associated with a decreased number of neuropeptide Y (NPY)+ cells in the medial prefrontal cortex (mPFC) and in the nucleus accumbens (NAc) shell and core subregions of pTHC rats compared to controls. Given the existing evidence, it is reasonable to speculate that the decrease in NPY signalling would significantly alter the functional activity of the projecting neurons (Vollmer et al., 2016), contributing to the cognitive impairment observed in pTHC offspring. Indeed, the decrease in the NPY-ergic tone observed in pTHC male adolescent offspring was associated with dampened limbic memory and instrumental learning, both functions dependent on the contribution of the mPFC (Caballero et al., 2019), and with salience attribution, regardless of the stimulus valence, a process dependent on NAc activity (Brown et al., 2000; Josselyn and Beninger, 1993; Warthen et al., 2019). The current study confirms the assumed hypothesis for cognitive impairment. Besides, a deeper investigation of NPY-ergic tone in the hippocampus is underway.

468      Although CB1R activation impairs memory formation, exposed mice were not treated with CB1 agonist before / during the test. What the authors found here is a long-term developmental consequence of THC exposure and not a direct effect of THC.

The working hypothesis is that the prenatal insult induced by THC may alter the neurodevelopmental trajectory, and, therefore, lead to a specific impairment in spatial cognitive processing in the offspring, through a perturbation in the effectors of eCBs-related excitatory/inhibitory balance in specific hippocampal neural circuits.

489      As a possible factor, which may contribute to the inferior performance of THC exposed rats attention deficit should be mentioned.

We cannot rule out that attention may be also affected. However. in this study, adolescent male offspring showed no deficits in spatial memory acquisition and object recognition memory on the first day. The following day, they did not remember the task. Furthermore, sustained attention and working memory are closely related functions that may share common mechanisms (J Neuropsychiatry Clin Neurosci. 2005 Summer;17(3):391-8). Thus, the parameter of working memory errors, which is described in the text, can be considered a measure of sustained attention. Since no impairment in working memory was observed, this evidence suggests that the effect of prenatal THC involves specific microcircuits for spatial- and object recognition reference memory in the adolescent male offspring. 

514      Previous studies found altered stress sensitivity (Lallai et al, 2022) and altered anxiety (Ellis et al, 2022) in exposed rats. The authors should discuss the reason of difference in their and previous findings.

We are thankful to the Reviewer for raising the point. The inconsistency with previous research that highlighted altered stress sensitivity and anxiety (Lallai et al, 2022; Ellis et al., 2022) shows that doses, time of exposure, route of administration, and experimental design may represent crucial factors in the identification of the functional consequences of pTHC exposure. Indeed, in our experimental conditions, THC exposure from PND 5 to 20 (a time frame resembling the first and second trimesters in humans) had no effect on emotional reactivity. Rather, in the study by Lallai et al. (2022), higher THC doses (5 mg/kg) administered orally from GD 1 to postnatal days to Wistar rats, induced altered behavioural reactivity and exploration. Then again, in the study by Ellis et al., (2022) intravenous, lower THC doses (0.15 mg/kg) from GD 5 to PND 2 had a main effect on the hedonic state in the adult male offspring.

518-527          The authors should compare their present findings in the hippocampus with their previous findings in the prefrontal cortex, amygdala and n.accumbens in the expression of target genes.

We highly appreciate the Reviewer’s suggestion. However, we believe it more appropriate to address the comparison in a differently structured manuscript, that has as its only, main focus the expression of the target genes in the corticomesolimbic regions. This is a great starting point for a valuable Review.

544      Using bulk expression analysis one can´t discriminate cell-line specific changes in the level of CB1 receptors, though CB1 activity on glutamatergic and GABAergic neurons has just an opposite effect on hippocampal activity. The authors should discuss this point.

We agree with the reviewer on the point raised. The two options have been discussed from lines 586 to 607. Given our data, we are not able to provide a detailed explanation, that would need anatomical and electrophysiological investigations. Given that in-utero exposure to CB1R agonists has been reported to significantly curtail the strength of the inhibitory signal by reducing the density of CB1R-containing inhibitory (CCK+) interneurons (CCK+-IN) [63], the increase in CB1R mRNA observed in our experimental conditions might provide an adaptive neuroplastic mechanism that scales specific CB1R+synapses, and balance the putative asymmetry between inhibitory and excitatory tone. On the other hand, HINT1 serves to enable the CB1R-NMDAR cross-regulation in the context of excessive NR1 expression. Thus, as a consequence of the increased expression of both CB1R and HINT1 here observed, an augmented removal of NR1 subunit from the cell membrane might result in glutamatergic hypo-function [33], thus promoting an increase in NR1 expression as a counter-adaptive mechanism. However, we may not rule out that other mechanistic dynamics can result from prenatal THC exposure, involving, for instance, astrocytes and mitochondria.

Round 2

Reviewer 1 Report

The Authors have not addressed my main concern. The present Manuscript does not contain any plasticity methodologies nor experiments, yet the Authors make several claims about the neuroplastic consequences of their manipulations. The expression of mGluR5, HINT1, CB1R, PSD95 and Homer1 mRNAs may impact some neuroplastic processes, but also a myriad of other phenomena. 

Author Response

Dear Reviewer 1,

Following your further comment, we have changed the discussion and conclusions, editing "neuroplasticity" in the text (highlighted in green). Indeed, you are right stating that we did not measure dynamic changes before and after the cognitive tasks in pTHC-exposed offspring. However, reasonably no one can deny that the molecular targets discussed in our manuscript are known to be main effectors of hippocampal neuroplasticity. 
